# Isotope and Spin Effects Induced by Compression of Paramagnetic Molecules

Irene Barashkova [1], Natalia Breslavskaya [1,2], Luybov Wasserman [1,3] and Anatoly Buchachenko [1,3,4,5,6,*]

1. Institute of Chemical Physics, Russian Academy of Sciences, 119991 Moscow, Russia
2. Institute of General and Inorganic Chemistry, Russian Academy of Sciences, 117907 Moscow, Russia
3. Institute of Biochemical Physics, Russian Academy of Sciences, 119991 Moscow, Russia
4. Institute of Problems of Chemical Physics, Russian Academy of Sciences, 142432 Chernogolovka, Russia
5. Scientific Center of the Russian Academy of Sciences, 142432 Chernogolovka, Russia
6. Chemical Department, Moscow State University, 119992 Moscow, Russia
* Correspondence: alb9397128@yandex.ru

**Abstract:** The zero-point energies (ZPEs) of paramagnetic molecules, free and compressed in a $C_{59}N$ paramagnetic cage, were computed. The excess of energy acquired by molecules under compression depended on the deuterium and tritium isotopes which ranged from 6–8 kcal/mol for $H_2^+$ to 1.0–1.5 kcal/mol for $HO^\bullet$ and $HO_2$. The differences in the ZPEs of compressed isotopic molecules resulted in large deuterium and tritium isotope effects which differed for singlet and triplet spin states. The hyperfine coupling (HFC) constants for protons and $^{17}O$ nuclei decreased under compression, confirming the leakage of the unpaired π-electron from the central oxygen atom of guest molecules into the system of π-electrons of the cage, and its distribution over 60 atoms of the $C_{59}N$. The latter seems to be the reason why the nitrogen-14 HFCs for $C_{59}N$ remain almost unchanged upon encapsulation of guest molecules. The singlet-triplet splitting is shown to depend on the Coulomb interaction, which controls the sign of the exchange potential. The importance of compression effects on the functioning of enzymes as molecular compressing devices is discussed.

**Keywords:** compressed molecules; zero-point energy; isotope effects; spin density

## 1. Introduction

The compression of molecules, as well as of atoms [1], is a means to modify their ionization potentials, electron affinities, and vibrational frequencies; the latter increase the differences in the zero-point energies of isotopic molecules, which results in increasing isotope effects. The effects induced by compression appear to be the only means of measuring molecular compression; they are of particular importance for understanding molecular machines—enzymes, which are known to stimulate enzymatic reactions by compression of reagents (particularly, in ATP and DNA synthesis). The purpose of this paper is to compute the zero-point energies (ZPEs) of isotopic paramagnetic ions and molecules, free and encapsulated in the paramagnetic $C_{59}N$ fullerene cage, and to estimate the compression-induced isotope and spin effects.

## 2. Calculation Procedure

Calculations of the energy characteristics, with full optimization of geometry, for all structures were performed using density functional theory (DFT) with an M06–2X functional and routine split valence basis set 6–31G* [2]. The M06-2X functional developed for calculating the non-covalent interactions was previously used with effect in studies of different complexes [3–6]. It should be noted that noncovalent interaction plays a major role in the stabilization of guest molecules inside fullerene. The calculations were carried out using the Gaussian 2009 program [7,8]. The results of vibration frequency calculations of all complexes characterized the optimized structures as the energy minima. Magnetic

parameters, hyperfine coupling constants (HFC) and g-factors for the optimized structures were calculated with the ORCA software package [9] using the B3LYP functional, together with the full-electron N07D basis set [10].

It is necessary to raise the extremely important question of whether, and to what extent, the calculations warrant confidence in them. In general, the calculations of the absolute magnitudes can be treated with less confidence than those dealing with differences in the magnitudes. The latter are deserving of more confidence because many uncertainties and errors are cancelled; this is particularly valid for isotopic molecules having identical chemical structures. The calculations of ZPE were carried out in different versions of the M06-2X functional, but the results were almost identical; all data presented below were obtained based on the M06-2X/6-31G* version.

## 3. Results and Discussion

### 3.1. Theory

The ZPE of the compressed molecule $\varepsilon_c$ is defined as the difference between the ZPE of the caged molecule and the ZPE of the cage itself; thus, for $HO^\bullet @C_{59}N$:

$$\varepsilon_c(HO^\bullet) = ZPE\ [HO^\bullet @C_{59}N] - ZPE\ [C_{59}N] \tag{1}$$

Similar equations are valid for all encapsulated molecules, as presented below (Table 1). The excess of compression energy $\Delta\varepsilon$, that is, the excess of ZPE acquired by compressed molecules under compression, is defined as the difference between the ZPE of compressed and free molecules:

$$\Delta\varepsilon = \varepsilon_c - \varepsilon_f \tag{2}$$

where $\varepsilon_f$ is the ZPE of the free molecule.

**Table 1.** ZPE of free and compressed molecules.

| Molecules | ZPE, kcal/mol | | |
|:---:|:---:|:---:|:---:|
| | Free | Compressed in S State | Compressed in T State |
| $H_2^+$ | 2.63 | 11.22 | 8.03 |
| $D_2^+$ | 1.86 | 8.45 | 5.24 |
| $T_2^+$ | 1.52 | 7.23 | 4 |
| $HO^\bullet$ | 5.3 | 6.81 | 7.18 |
| $DO^\bullet$ | 3.86 | 5.15 | 5.35 |
| $TO^\bullet$ | 3.24 | 4.48 | 4.55 |
| $HO_2^\bullet$ | 9.18 | 10.57 | - |
| $DO_2^\bullet$ | 7.21 | 8.47 | - |
| $TO_2^\bullet$ | 6.38 | 7.56 | - |
| $O_2^-$ | 1.89 | 1.71 | 1.99 |

The compression-induced deuterium isotope effect on ZPE of $HO^\bullet$ and $DO^\bullet$ is determined by the difference:

$$\varepsilon_c^* = \varepsilon_c\ (HO^\bullet) - \varepsilon_c\ (DO^\bullet) \tag{3}$$

Similarly, the deuterium isotope effect on ZPE of free molecules is determined by the difference:

$$\varepsilon_f^* = \varepsilon_f\ (HO^\bullet) - \varepsilon_f\ (DO^\bullet) \tag{4}$$

Deuterium isotope effects on the dissociation rate constants of compressed and free molecules are given by the ratios:

$$(k^H/k^D)_c = \exp\ (\varepsilon_c^*/RT) \tag{5}$$

and

$$(k^H/k^D)_f = \exp\ (\varepsilon_f^*/RT) \tag{6}$$

respectively. The identical equations are also valid for tritium molecules and tritium isotope effects; they are presented in Tables 2 and 3.

**Table 2.** Deuterium isotope effects on free and compressed molecules.

| Isotopic Pair | Free | Compressed in S State | Compressed in T State |
|:---:|:---:|:---:|:---:|
| $H_2^+$ vs. $D_2^+$ | 3.2 | - | - |
| $H_2^+$ @$C_{59}$N vs. $D_2^+$ @$C_{59}$N | - | 101.2 | 104.6 |
| $HO^\bullet$ vs. $DO^\bullet$ | 11 | - | - |
| $HO^\bullet$ @$C_{59}$N vs. $DO^\bullet$ @$C_{59}$N | - | 15.9 | 21.1 |
| $HO_2^\bullet$ vs. $DO_2^\bullet$ | 26.7 | - | - |
| $HO_2^\bullet$ @$C_{59}$N vs. $DO_2^\bullet$ @$C_{59}$N | - | 33.1 | - |

**Table 3.** Tritium isotope effects on free and compressed molecules.

| Isotopic Pair | Free | Compressed in S State | Compressed in T State |
|:---:|:---:|:---:|:---:|
| $H_2^+$ vs. $T_2^+$ | 6.4 | - | - |
| $H_2^+$@$C_{59}$N vs. $T_2^+$ @$C_{59}$N | - | 772.8 | 826.1 |
| $HO^\bullet$ vs. $TO^\bullet$ | 31 | - | - |
| $HO^\bullet$@$C_{59}$N vs. $TO^\bullet$ @$C_{59}$N | - | 48.6 | 80.1 |
| $HO_2^\bullet$ vs. $TO_2^\bullet$ | 106.3 | - | - |
| $HO_2^\bullet$@$C_{59}$N vs. $TO_2^\bullet$ @$C_{59}$N | - | 150.9 | - |

### 3.2. Zero-Point Energies

A paramagnetic molecule encapsulated in $C_{59}$N forms a two-spin system, which may be in two spin states, singlet and triplet; both are presented in Table 1. The table summarizes the computed ZPEs of the free and compressed molecules. Evidently, the compression of molecules increases their ZPEs; the excess of compression energy $\Delta\varepsilon$ acquired by a molecule under compression is quite significant, about 2–3 kcal/mol for $HO^\bullet$ and $HO_2^\bullet$, and even larger, about 8 kcal/mol, for $H_2^+$. For $HO^\bullet$, the ZPE in the triplet state slightly increases with respect to that in singlet state, while for $H_2^+$, in contrast, it decreases; the reason appears to be an attraction between the positive charge of the central ion and the local negative charge of the cage induced by polarization. In the sequence, $HO^\bullet$, $HO^\bullet$@$C_{60}$, $HO^\bullet$@$C_{59}$N (singlet), and $HO^\bullet$@$C_{59}$N (triplet), the ZPEs increase only slightly to 5.30, 6.60, 6.81 and 7.18 kcal/mol.

### 3.3. Isotope Effects

The isotope effects were calculated according to Equations (3)–(6) and are presented in Tables 2 and 3. The ratios $k^H/k^D$ of the rate constants for the dissociation of covalent bonds H–H and O–H and their deuterium analogues are given in Table 2. Thus, for the pair $H_2^+$–$D_2^+$, $k^H/k^D$ is denoted as $H_2^+$ vs. $D_2^+$, and, for the free molecules, it is equal to 3.2; however, for the pair of compressed molecules $H_2^+$@$C_{59}$N vs. $D_2^+$@$C_{59}$N, it increases to the values 101.4 and 104.6 for the singlet and triplet states, respectively. Similarly, the $k^H/k^D$ values for the $HO^\bullet$ and $DO^\bullet$ radicals increase from 11.0 for the free radicals to 15.9 and 21.1 for the singlet and triplet states, respectively. For the pair $HO_2^\bullet$–$DO_2^\bullet$, the compression-induced isotope effect is smaller, varying from 26.7 to 33.1 (Table 2).

Evidently, the measurement of isotope effects represents a tool for testing molecular compression. However, both the ZPE and isotope effects only confirm the supply of energy stored in compressed molecules; they cannot be used in closed cages, such as fullerenes, but they may be implemented in partly open devices, such as enzymes. This may be significant for understanding abnormally large isotope effects in enzymatic reactions, where they are attributed to tunneling, but such effects may be caused by the compression of molecules in the catalytic sites of enzymes. This alternative may be significant for elucidating the physics of enzymatic reactions [11].

Tritium isotope effects, as shown in Table 3, are also sensitive to compression. Thus, for the pair $H_2^+$–$T_2^+$, $k^H/k^T$ is denoted as $H_2^+$ vs. $T_2^+$, and, for the free molecules, is equal to 6.4; however, for the pair of compressed molecules, $H_2^+@C_{59}N$ vs. $T_2^+@C_{59}N$, it increases to values of 772.8 and 826.1 for the singlet and triplet states, respectively. Similarly, the value of $k^H/k^T$ for $HO^\bullet$ and $TO^\bullet$ increases from 31.0 for the free radicals, to 48.6 and 80.1 for the compressed singlet and triplet states, respectively. For the pair $HO_2^\bullet$–$TO_2^\bullet$, the compression-induced isotope effect is smaller: it varies from 106.3 to 150.9 (Table 3). It is worth noting that the difference in compression-induced isotope effects, for both deuterium and tritium, in singlet and triplet states, is not too large.

### 3.4. Singlet-Triplet Splitting

For $HO^\bullet@C_{59}N$, $HO_2^\bullet@C_{59}N$, and $O_2^-@C_{59}N$ molecules, the lowest energy spin state is a triplet; in this sequence, the singlet is higher than the triplet by 2.8, 17.4 and 21.4 kcal/mol, respectively. In contrast, for $H_2^+@C_{59}N$, the ground state is a singlet; the triplet state is above that of the singlet by 43.2 kcal/mol. This anomaly reproduces the similar anomaly in ZPE for $H_2^+@C_{59}N$ (see above); both are seemingly induced by strong Coulomb interaction between the positive charge of the central ion and the local negative charge on the cage, induced by polarization. This unusual situation occurs when Coulomb interaction controls the sign of the exchange potential. Evidently, the strong Coulomb interaction modifies the spin density distribution, violating the exchange potential.

### 3.5. Spin Densities

The nitrogen-14 HFCs for $C_{59}N$ remain almost unchanged upon encapsulation of guest molecules, but the HFCs of captured molecules are highly sensitive to the compression; they are shown in Table 4.

**Table 4.** Hyperfine coupling constants (G).

| Molecules | $a$ (H) | $a$ ($^{17}O$) |
|:---:|:---:|:---:|
| $HO^\bullet$ | −23.8 * | −38.8 |
| $HO^\bullet@C_{59}N$ triplet | −9.7 | −15.6 |
| $O_2^-$ | - | −25.0 |
| $O_2^-@C_{59}N$ triplet | - | −23.3 |

* It is close to −27G measured experimentally [12].

The HFC constants $a$ (H) for protons in the $HO^\bullet$ radical, both free and captured by $C_{59}N$, are negative; this implies that the spin density on the proton arises by spin polarization of the O–H σ-bond by the unpaired π-electron on the oxygen atom [13]. The proton HFC constants indicate that about 5% of the unpaired electron penetrates the *s*-orbital of H in free OH and only 2% in the compressed OH. The spin density on the $^{17}O$ nuclei is positive, but the HFC constants $a(^{17}O)$ are negative because the nuclear magnetic moment of $^{17}O$ is negative. For the same reason, the positive spin density on the oxygen atoms in $O_2^-$ induces negative HFC constants on the $^{17}O$ nuclei. Figure 1 demonstrates schematically the position of $HO^\bullet$ in the cage.

It is remarkable that the HFCs in the $HO^\bullet$ radical decrease almost identically for both hydrogen and oxygen atoms under compression, by 2.5 times. This confirms that there is leakage of the unpaired π-electron from the central oxygen atom into the system of π-electrons of the cage and the distribution of transferred spin density on the 60 atoms of $C_{59}N$. It appears to be the reason why the nitrogen-14 HFC constants for $C_{59}N$ remain almost unchanged upon encapsulation of guest molecules. A similar effect is evident for the ion $O_2^-$ but, in this case, the decrease in HFC is much less, amounting to only 20%. The changes in the g-factors of the guest molecules induced by their encapsulation were computed to be negligible.

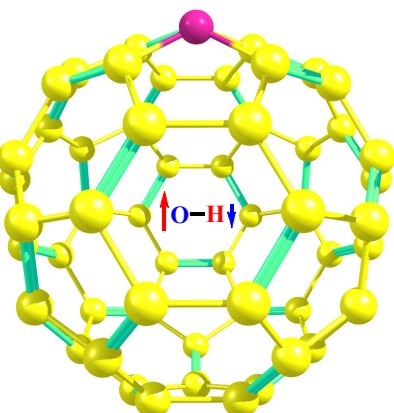

**Figure 1.** HO• radical encapsulated into $C_{59}N$. The large positive spin density on the oxygen atom is indicated by the red arrow; the small negative spin density on the hydrogen atom is shown by a blue, inversely oriented arrow.

*3.6. Applicability to Enzymes*

Enzymes, as molecular compressing devices, are extraordinary targets; they are known to be powerful molecular motors (rotors, as in ATP synthase, or linear pumps, as in kinases), in which molecules are confined and compressed by non-covalent interactions. Their power can be measured quantitatively as a force, by which the enzyme continues to synthesize biomolecules. For this purpose, single molecule technologies have been developed, the key units of which are optical traps or optical tweezers, the latter being formed of polystyrene or quartz bead. On exposure to a focused laser beam, the bead is drawn into the focus with a force which depends on the light intensity. Hence, the trap attached to the end of a molecule pulls the molecule with a force, which can be measured. This technique was used to study molecular motors (ATP synthases, kinesin, myosin, immunoglobulin proteins, etc.) and their functioning [14–17].

Thus, RNA polymerase draws a DNA double strand inside the enzymatic site and pulls it, simultaneously generating an RNA chain, with a force of 25 pN. This is considered the most powerful among the known molecular motors, being 5–6 times more powerful than kinesin. It is assumed that most of the energy of this motor is spent on unwinding the DNA strands [15]. Another protein motor is *l*-exonuclease, a part of bacterial viruses. Unlike RNA polymerase, which is powered by ATP, *l*-exonuclease operates on the DNA energy: while pulling the DNA double strand through itself, it "eats up" one of the strands. It is a weaker motor than RNA polymerase and develops a force of only 5 pN [17].

In chemical reactions the classical, mass-dependent, isotope effects arise solely from the differences in the zero-point energies of isotopic species; for this reason, the effects are unaffected by pressures which are not large enough to violate the inter-molecular potentials. It is an unambiguous indicator, discriminating the reaction mechanisms, whether classical or tunneling. The latter is known to strongly depend on the molecular motion, which modulates the height and the profile of the activation barriers controlling their quantum penetrability. Thus, Northrop et al [18,19] found neither deuterium nor $^{13}C$ pressure induced isotope effects on the oxidation reaction (shown below) of benzyl alcohol by yeast alcohol dehydrogenase YADH, confirming that this hydride ion transfer reaction followed a classical mechanism. Indeed, the fact that the decrease in activation volumes for hydride transfer was equivalent when one mass unit was added to the carbon end ($^{13}C$) of a scissile C–H bond, and when one mass unit was added to the hydrogen end (deuterium), excludes tunneling as the mechanism (Figure 2).

**Figure 2.** The scheme of the hydride ion transfer reaction.

However, in enzymatic reactions of hydrogen (deuterium) atom transfer in the oxidation of linoleic acid by soybean lipoxygenase, a giant isotope effect of $k^H/k^D \approx 80$ was observed [20,21]; for mutant lipoxygenase, the effect $k^H/k^D$ reached 500–700 [22,23]. The discovery of this anomalous isotope effect is a significant event in chemistry; the effect is intriguing as it was observed under mild conditions, in which tunneling seems to be inconceivable. The isotope effect depends on the solvents ($H_2O$ and $D_2O$), and the temperature and concentration of the substrate. The most remarkable feature is that the isotope effect depends on the enzyme mutant forms; the modification of the catalytic site by mutagenesis significantly changes both the catalytic activity of the enzyme and the isotope effect. The latter allows consideration of the idea that the isotope anomalies may be induced by molecular compression in the enzymatic pocket of reagents, modifying the ZPE of their C–H (and C–D) bonds and their chemical reactivity in hydrogen atom transfer.

Recently, it was found that some forms of nitrogenases—the vanadium- and iron-only nitrogenases—produce methane from carbon dioxide and water molecules [24]. Remarkably, the nitrogenase-derived methane is depleted with deuterium; the isotope effects, of 2.071 and 2.078, respectively, are not too large, but they are notably larger than those for all other known methane-producing enzymes. Of course, the effects cannot be attributed for certain to the individual reaction; however, it is not excluded that this anomaly stems from the compression of the enzymatic site in these sorts of nitrogenases.

It is of note that molecular compression induces new magnetic catalysis of enzymatic reactions. ATP and DNA synthesis are generally accepted to occur as nucleophilic reactions, which implies that the reaction proceeds as an addition of phosphate to the ADP molecule (ATP synthesis) and an attachment of the nucleotide to the terminal ribose ring of the growing DNA chain (DNA synthesis). However, in order to accomplish a nucleophilic reaction, it is necessary to overcome powerful repulsion between reactants with closed electronic shells. The nucleophilic transfer of the phosphate group is strongly energy deficient and was demonstrated by reliable calculations of the energy barriers to require 42–46 kcal/mole [25]. Similarly, nucleophile attack of the hydroxyl group to hydrolyze ATP needs to overcome an energy barrier of 39 kcal/mole [26]. The high barrier for the nucleophilic reaction is the reason why ATP and DNA syntheses are accomplished only by ATP synthase and DNA polymerases, which are special molecular machines. The source of energy needed to cover the energy deficit for energy-expensive phosphorylation is thought to be compression of the catalytic site, induced by protein mechanical energy, which compresses the reactants to overcome their repulsion.

These syntheses are known to be catalyzed by $Zn^{2+}$, $Ca^{2+}$ and $Mg^{2+}$ ions; the ions were traditionally considered to coordinate reactants in the catalytic site by keeping them on the reaction trajectory to facilitate nucleophilic attack, and to probably slightly modify their reactivity due to the partial redistribution of charges in the reactants. However, the substitution in the enzyme sites of these ions by the ions $^{25}Mg^{2+}$, $^{67}Zn^{2+}$, and $^{43}Ca^{2+}$ with magnetic nuclei was shown to produce huge isotope effects; with respect to the nonmagnetic ions, these nuclear magnetic ions stimulated ATP synthesis by 3–5 times [27–29], and inhibited DNA synthesis by 3–5 times [30–32].

The observation of magnetic effects is unambiguous evidence that the ATP and DNA syntheses occur through a magnetically sensitive radical pair mechanism, which implies electron transfer from the reagent to the metal ion in the catalytic site; the mechanisms of these reactions were suggested and substantiated in [32,33].The radical pair mechanism is induced by the remarkable capacity of enzymes to squeeze water molecules out of the catalytic site [34,35], when the enzyme domains are drawn together to unite reagents. The

compression of the site partly dehydrates the catalyzing ions M $(H_2O)_n^{2+}$ (M is Mg, Zn, or Ca), increasing both the positive charge on the core metal and the electron affinity of the ion, so that, at some threshold value $n^*$, electron transfer becomes exoergic and energy allowed. The water molecule with number $n^*$ in the complex M $(H_2O)_n^{2+}$ functions as a trigger—it switches over the reaction between endoergic and exoergic regimes. At $n > n^*$, electron transfer is endoergic, while, at $n < n^*$, it is exoergic and energy allowed. When $n$ reaches $n^*$, electron transfer becomes inevitable, switching on the new radical pair mechanism [36].

The radical pair mechanism, induced by compression of molecules in the enzymatic site and discovered using nuclear magnetic isotopic ions, is a unique means to accomplish magnetic control; it elucidates how to stimulate ATP synthesis and eliminate ATP deficiency in hypoxia and cardiac diseases, how to use magnetic isotope ions as a means for controlling gene expression and cell proliferation, and how to stimulate the destruction and apoptosis of cancer cells [33,37–39].

## 4. Conclusions

Isotope effects, induced by molecular compression, are suggested as a probe for testing the molecular compression of enzymatic sites; they may be important for elucidating the physics of enzymes as molecular machines, particularly for understanding the enormously large isotope effects observed in some enzymatic reactions.

**Author Contributions:** Data curation, I.B.; Formal analysis, N.B., L.W. and A.B. All authors have read and agreed to the published version of the manuscript.

**Funding:** This research received no external funding.

**Institutional Review Board Statement:** Not applicable.

**Informed Consent Statement:** Not applicable.

**Data Availability Statement:** Not applicable.

**Conflicts of Interest:** The authors declare no conflict of interest.

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
