# Peer review of "Isotope and Spin Effects Induced by Compression of Paramagnetic Molecules"

_2673-7167, doi:10.3390/physchem2030018_

Round 1

Reviewer 1 Report

Barashkova et al present a computational study of isotope effects H2+, HO. and HO2. in a C59N cage. Their conclusions are: "The effects, induced by molecular compression, as a new source of isotope effects are suggested as a probe for testing molecular compression of enzymatic sites; they may be important for elucidating physics of enzymes as the molecular machines, particularly for understanding enormously large isotope effects observed in some enzymatic reactions."

The authors need to define "molecular compression". There have been a number of groups studying the effect of hydrostatic pressure of isotope effects on enzyme reactions. Some of these studies are noted in the manuscript. It's then now clear what the the "new source" of isotope effects is. If it's the encapuslation, it's odd that most of the discussion focuses on enzyme effects. There is a footnote that the HO. hyperfine coupling is similar to the experimental value, but there is no other benchmarking of the calculations, nor any description of how the calculations were set up -  where did the coordinates of the cage come from? How did the authors check their geometries were in a global minimum, etc?

It's also not clear why the authors chose to use the C59N cage, and the particular isotopic molecules - H2+, HO. and HO2. How do these relate to enzymes and did the authors consider computing bond dissociation energies relevant to the generation of these molecules?

The manuscript is reasonably well organised, but should make it clear the enzyme content is "Discussion". The standard of English is not good and the manuscript needs quite extensive proof reading.

Reviewer 2 Report

The authors report a pure computational study on the physicochemical properties of a paramagnetic molecule encapsulated in a paramagnetic cage.

The whole study is developed with the aid of DFT theory.

One issue, of major importance, is whether the selected functional and basis are appropriate in order to account for the concept. My feeling is that 6-31G is not enough for such a study, in a highly charge transfer systems and whereas electron compression is important, the augmentation with the diffuse and/or polarization functions seems to be a meaningful choice. The author could try to augment 6-31G and to study several trends and alterations. XC  functional could also be of importance, since electron correlation effects are dominant. You could try as a reference CCSD method, with a medium size basis. M06-2X is a good choice for the study of such systems, although the use of a LR functional (e.g CAM_B3LYP, LCBLYP) could reveal the CT effect.

It would be nice to depict the spin electron distribution and to highlight on difference of spin density. This is possible with GAUSSIAN.

Round 2

Reviewer 2 Report

Although the authors did not consider in detail the comments raised by the reviewer, the article improved is some points.

Prior to any acceptance the authors should  comment on the suitability of the used functional.

Author Response

Author’s comment to the Review R2

Prior to any acceptance the authors should comment on the suitability of the used functional.

Frankly, I didn’t understand this comment… All calculations were carried out in terms of the well-known and generally accepted functional for energy, optimization of geometry, and for HFC computing. Their suitability and reliability were many times substantiated and described in hundreds of papers and books, cited partly in the revised MS. No needs to reproduce them. Moreover, we controlled reliability of the functional for HFC constants (Hyper Fine Coupling constants) by calculation of the a(H) for HO radical. I ask my respectable Reviewer to accept my arguments.

ALB, the corresponding author.
